# Comparative Effect of Two Types of Physical Exercise for the Improvement of Exercise Capacity, Diastolic Function, Endothelial Function and Arterial Stiffness in Participants with Heart Failure with Preserved Ejection Fraction (ExIC-FEp Study): Protocol for a Randomized Controlled Trial

**DOI:** 10.3390/jcm12103535

**Published:** 2023-05-18

**Authors:** Iván Cavero-Redondo, Alicia Saz-Lara, Irene Martínez-García, Bruno Bizzozero-Peroni, Valentina Díaz-Goñi, Ana Díez-Fernández, Nerea Moreno-Herráiz, Carlos Pascual-Morena

**Affiliations:** 1Facultad de Ciencias de la Salud, Universidad Autónoma de Chile, Talca 3460000, Chile; 2Health and Social Research Centre, Universidad de Castilla-La Mancha, 16001 Cuenca, Spain; 3Instituto Superior de Educación Física, Universidad de la República, Rivera 40000, Uruguay; 4Facultad de Enfermería, Universidad de Castilla-La Mancha, 16001 Cuenca, Spain

**Keywords:** physical exercise, heart failure, exercise capacity, diastolic function, endothelial function, arterial stiffness

## Abstract

(1) Background: Heart failure (HF) with preserved ejection fraction (HFpEF) accounts for approximately 50% of all patients with HF. In the absence of pharmacological treatments that have been successful in reducing mortality or morbidity in this pathology, physical exercise is recognized as an important adjunct in the treatment of HF. Therefore, the objective of this study is to compare the efficacy of combined training and high intensity interval training (HIIT) on exercise capacity, diastolic function, endothelial function, and arterial stiffness in participants with HFpEF. (2) Methods: The ExIC-FEp study will be a single-blind, 3-arm, randomized clinical trial (RCT) conducted at the Health and Social Research Center of the University of Castilla-La Mancha. Participants with HFpEF will be randomly assigned (1:1:1) to the combined exercise, HIIT or control group to evaluate the efficacy of physical exercise programs on exercise capacity, diastolic function, endothelial function, and arterial stiffness. All participants will be examined at baseline, at three months and at six months. (3) Results: The findings of this study will be published in a peer-reviewed journal. (4) Conclusions: This RCT will represent a significant advance in the available scientific evidence on the efficacy of physical exercise in the treatment of HFpEF.

## 1. Introduction

Heart failure (HF) is a chronic disease with a major and increasingly serious social and health impact. It currently affects more than 23 million people worldwide, and its prevalence is expected to increase by approximately 25% by 2030 [1]. Two distinct phenotypes of HF have been identified: HF with reduced ejection fraction (HFrEF) and HF with preserved ejection fraction (HFpEF) [2]. In the last decade, HFpEF has accounted for approximately 50% of all patients with HF. In general, the prognosis of HFpEF is slightly better than the prognosis of HFrEF [3,4]. The increasing prevalence of HFpEF is attributed to increased life expectancy, better awareness of the diagnosis [5] and increased prevalence of major underlying risk factors (e.g., obesity, diabetes mellitus and hypertension) [6].

Insufficient physical activity, poor cardiorespiratory fitness and obesity have been identified as significant risk factors for cardiovascular disease (CVD) in general and HF in particular, mainly in patients with HFpEF [7,8,9,10]. HFpEF is a clinical syndrome in which the heart is unable to supply the amount of oxygen required by the tissues according to their needs or does so only at the cost of excessively increased left ventricular filling pressures, despite the existence of a left ventricular ejection fraction (LVEF) within the normal range. In the absence of pharmacological treatments that have been successful in reducing the mortality or morbidity of this pathology, the therapeutic goal for these patients is based on improving quality of life (QoL) by improving exercise capacity. Physical exercise is recognized as an important adjunct in the treatment of HF and is recommended by the American College of Cardiology/American Heart Association (ACC/AHA) and European Society of Cardiology (ESC) guidelines [11,12]. Currently, aerobic exercise is the most studied physical exercise in this population, but in recent years, the combination of aerobic exercise with strength training (combined exercise (CE)) and high intensity interval training (HIIT) have emerged.

To date, several systematic reviews and meta-analyses of randomized clinical trials (RCTs) have suggested that physical exercise can lead to important changes in different parameters of exercise capacity, such as fitness as measured by VO2max and diastolic function as measured by echocardiographic parameters [13,14,15,16,17,18,19,20,21]. However, most of these systematic reviews and meta-analyses analyzed all types of exercise together and did not explore the effect on endothelial function and arterial stiffness. In addition, there are currently eleven published RCTs [22,23,24,25,26,27,28,29,30,31,32] on the effect of physical exercise on different parameters of exercise capacity, mainly VO2max. Most of these compare the effect of moderate aerobic exercise versus a control group, while two clinical trials and one pilot study compare moderate aerobic exercise versus HIIT (with one of these studies including a control group) and a single pilot study comparing CE versus a control group. Only two studies analyzed the effect of moderate aerobic exercise on endothelial function and arterial stiffness [28,29]. Recently, a network meta-analysis compared the effect of aerobic exercise, CE and HIIT, supporting the well-known effect of aerobic exercise but showing promising findings for CE and HIIT not only on VO2max but also on diastolic function [33].

Due to the lack of studies analyzing the effect of CE and HIIT and their impact, not only on improving exercise capacity and cardiac function but also on endothelial function and arterial stiffness, in participants with HFpEF, this RCT will be conducted to (1) compare the efficacy of CE and HIIT on exercise capacity, diastolic function, endothelial function and arterial stiffness in patients with HFpEF and (2) to compare the efficacy of CE and HIIT on QoL in participants with HFpEF.

## 2. Materials and Methods

### 2.1. Design

The ExIC-FEp study will be a single-blind, three-arm RCT to analyze two types of supervised physical exercise in participants with HFpEF for 6 months. Additionally, this protocol for RCT has been registered for clinical trials (NCT05726474).

### 2.2. Setting

The recruitment of participants with HFpEF will be carried out in the Cuenca I, II and III primary health centers and the cardiology and internal medicine services of the Hospital Virgen de la Luz of the Castilla-La Mancha Health Service (SESCAM). The professionals of these services will be informed of the RCT and its purpose to recruit patients with HFpEF, who will then be invited via these healthcare professionals, as well as information meetings and posters, to participate in the RCT.

### 2.3. Participants 

The study will include sedentary participants with stable symptomatic HFpEF, diagnosed according to ESC criteria [34]. Participants with HFrEF will be excluded, as well as patients with other causes of HF symptoms, such as significant valvular or coronary heart disease or uncontrolled hypertension or arrhythmias. The inclusion and exclusion criteria are shown in Table 1.

For inclusion and exclusion criteria, a complete echocardiographic examination incorporating all two-dimensional and Doppler data will be performed. In addition, the patient’s clinical assessment, exercise capacity and medical history will be taken into account and spirometry will be performed to rule out significant pulmonary disease.

### 2.4. Randomization

After informed consent and screening, participants will be randomly distributed (1:1:1) using EPIDAT 4.2 software (Figure 1) into three groups: group 1, CE (continuous aerobic exercise with moderate intensity strength exercises); group 2, HIIT; and group 3, control group (CG).

### 2.5. Intervention

#### 2.5.1. Training Interventions

Participants in groups 1 and 2 will perform supervised intensity training on an ergometer bicycle. Exercise intensity (relative to maximum heart rate (HRmax) and maximum heart rate percentage (%HRmax) will be determined by a maximal cardiopulmonary stress test at baseline. 

**CE** (Group 1). Participants will exercise for 40 min three times a week on an ergometer bike at 50–60% of VO2max, 60–70% of HRmax and 11–13 on the Borg scale, without respiratory distress. In addition, strength training (bench press, leg press, leg curl, rowing machine, triceps dip, pec pull) will be performed twice a week. The strength training will be performed with 15 repetitions per exercise per session, with a workload corresponding to 60% to 65% of the 1 repetition maximum (1RM) measured at the beginning and end of the intervention.

**HIIT** (Group 2). Participants will perform three training sessions per week. Each training session will begin with a 10-min warm-up at moderate intensity (corresponding to 50–60% of VO2max, 60–70% of HRmax, and 11–13 on the Borg scale, without respiratory distress) before four 4-min intervals at high intensity (corresponding to 85–90% of VO2max, 90–95% of HRmax, and 15–17 on the Borg scale, with mild respiratory distress). Each interval will be separated by 3 min of active pauses, with an HRmax of 50–70%. The training session will end with 3 min of cool down at moderate intensity (corresponding to 50–60% of VO2max, 60–70% of HRmax, and 11–13 on the Borg scale, without respiratory distress). The total exercise time will be 40 min for the HIIT group. 

#### 2.5.2. Control Group (Group 3)

CG participants will receive standard advice on the general cardiovascular benefits of physical exercise according to current guidelines (30 min of walking on most days of the week) [35] and will continue their usual daily leisure-time physical activity throughout the study.

#### 2.5.3. Duration of the Intervention

Interventions will last 3 months. Figure 2 shows the flow chart for participant inclusion and follow-up: screening, inclusion and exclusion, randomization, baseline testing, and testing after follow-up. 

### 2.6. Criteria for Nonadherence and Prolongation of Training 

For participants to be considered compliant with training, they must attend at least 70% of the possible training sessions within the allotted time. Training compliance will be monitored and documented regularly throughout the trial. If, due to unforeseen circumstances, a participant does not comply with training during the supervised intervention (6 months), supervised training may be extended for up to 4 weeks.

### 2.7. Outcomes

#### 2.7.1. Primary Outcomes


*Exercise capacity by maximal cardiopulmonary exercise test with Ergoline600 ergometer bicycle:*
−Gas analyzer (K5 COSMED): VO2max, ventilation slope/carbon dioxide production (VE/VCO2) per minute, and ventilatory threshold (VT).−Workload measured in watts.−Average exercise time in minutes.−HRmax by Polar H10 heart rate monitor.−A 6-min walking test, in a 30-m corridor, to be performed by walking from one side to the other along this stretch of corridor, which will be delimited by cone-type indicators. These signs will be placed 29 m from each other, leaving 0.5 m at each end for the subject to turn. The test will be performed accompanied by the examiner. The primary test outcome is the final distance walked.



*Echocardiography:*
−Using Sonosite SII Doppler ultrasound (Sonosite Inc., Bothell, WA, USA), the following will be measured: E velocity (m/s), A velocity (m/s), E/A ratio, e’ velocity (m/s), E/e’ ratio, ejection fraction percentage, left ventricular volume index, end-diastolic volume, left ventricular mass, left atrial diameter, isovolume relaxation time, deceleration time (m/s) and left atrial volume index.



*Endothelial function:*
−Carotid intima-media thickness (cIMT): by ultrasound with the Sonosite SII device (Sonosite Inc., Bothell, WA, USA).



*Arterial stiffness:*
−Pulse wave velocity (PWv) and central augmentation index (cAIx) using the Mobil-O-Graph PWA (I.E.M. GmbH, Stolberg, Germany).



*Quality of life:*
−Validated 12-item health questionnaire (SF-12).−Validated Minnesota Living with Heart Failure Questionnaire (MLWHFQ).


#### 2.7.2. Covariables


*Sociodemographic variables:*
−Age.−Sex.−Socioeconomic level: using the Spanish Society of Epidemiology scale. Participants will report their educational level and employment status, and an index will be calculated considering both.



*Patient medical history:*
−Comorbidities (hypertension, diabetes, smoking, alcohol consumption).−Pharmacological treatment.



*Anthropometric variables:*
−Weight: mean of two weight measurements (Seca^®^ 861 scale), with the participant barefoot and lightly wrapped.−Height: mean of two measurements with a wall-mounted measuring rod (Seca^®^ 222), with the participant barefoot, in an upright position and with his/her sagittal midline coinciding with the measuring rod.−Body mass index: body mass index will be calculated by the formula weight (kg)/height^2^ (m^2^).−Waist circumference: mean of two measurements of waist circumference with a flexible tape measure at the midpoint between the last rib and the iliac crest at the end of a normal exhalation.−Body fat percentage: mean of two measurements from an 8-electrode electrical bioimpedance model Tanita^®^ BC-418 MA (Tanita Corp., Tokyo, Japan).−Densitometry (DXA) (model GE-Lunar DXA): densitometry will be performed to estimate fat mass, lean mass and bone mineral density.−Blood pressure: mean of two blood pressure determinations separated by a 5-min interval. The first measurement is obtained after a rest of at least 5 min. The subject is seated in a quiet environment, with the right arm semi-flexed at heart level. Blood pressure will be measured with the Omron^®^ M5-I monitor (Omron Healthcare UK Ltd., Milton Keynes, UK) and cuffs of three sizes according to arm circumference.



*Muscle strength:*
−The manual grip strength will be determined with the TKK 5401 Grip-D dynamometer (Takey^®^, Tokyo, Japan).



*Spirometry (Datospir Touch Easy-T):*
−Forced vital capacity (FVC).−Forced expiratory volume in the first second (FEV1).−FEV1/FVC ratio.



*Biochemical parameters:*


Blood will be drawn from a vein in the antecubital fossa between 8:15 and 9:00 a.m., after at least 12 h of fasting.

The following will be determined:
−Glucose, total cholesterol, triglycerides, HDL cholesterol, LDL cholesterol, apolipoproteins A1 and B, insulin and ultrasensitive C-reactive protein. The determinations will be made on a Roche Diagnostics^®^ Cobas 8000 system, and the insulin determination will be made on the Abbott^®^ Architect platform.−N-terminal pro-B-type natriuretic peptide (NT-proBNP). Plasma concentrations of NTproBMP will be determined by enzyme-linked immunosorbent assay (ELISA) (R & D Systems, Minneapolis, MN, USA).−HbA1c will be determined by high-performance liquid chromatography on an ADAMS A1c HA-8180 V analyzer from A. Menarini Diagnostics^®^, a method certified by the National Glycohemoglobin Standard Glycohemoglobin Standardization Program (NGSP) and the International Federation of Clinical Chemistry and Laboratory Medicine (IFCC).


*Physical activity:*
−Accelerometry: Physical activity will be objectively measured using AX6 accelerometers (Axivity) for nine consecutive days (including nights).−Podometry measured by using the Xiaomi MI Band 3 smart bracelet. This device will also be used to provide information during the study on sleep time and sedentary time.



*Mediterranean diet adherence:*
−Validated 14-item Mediterranean diet adherence questionnaire (MEDAS-14).


The basic clinical features are shown in Appendix A.

### 2.8. Ethical Considerations

This study protocol has been approved by the Clinical Research Ethics Committee of the Hospital Virgen de la Luz in the city of Cuenca where the study is to be performed (REG: 2022/PI2122).

Participants will be informed of the objectives and methods of the study and will be asked to confirm in writing their approval to participate in the project.

### 2.9. Sample Size Calculation

The sample size calculation is based on data from the RCT by Mueller et al. [31] which reveals that 21 participants per group would provide 80% power at *p* < 0.05 to detect a statistically significant result for VO2max, the primary outcome. Considering the three groups (CE, HIIT and GC) and a dropout rate of 20%, a total of 72 participants will be recruited.

### 2.10. Statistical Analysis

The statistical analysis will consist of three phases. The first will consist of testing the effectiveness of randomization in creating three comparable groups of participants with HFpEF, exploring the presence of outliers, and testing the degree of adjustment of the main variables to the normal distribution.

In the second phase, ANCOVA models will be used (baseline and 3 months) in which the dependent variables will be each outcome variable, and the intervention will be the fixed effect (2 = CE, 1 = HIIT and 0 = GC), adjusted for baseline, age, sex and socioeconomic status. The results will be expressed as absolute differences in changes in variables between baseline and final measurements (95% confidence interval (95% CI)). Bonferroni post hoc tests will be performed for paired comparisons.

In the third phase, a comparison of CE, HIIT and CG will be performed using the propensity score matching method to take into account covariate imbalance in baseline measurements between subjects. The propensity score estimates the effect of the intervention using a causal inference model, which explains what would have happened if all subjects in the intervention and control groups had the same characteristics at baseline. Each subject is matched to a subject with similar characteristics using a 0.40 caliper using the psmatch2 command.

All analyses will be performed by INTENT TO TREAT, keeping subjects in the originally assigned intervention or control group, regardless of the number of physical activity program sessions attended, and considering the CONSORT.

The results with *p* < 0.05 will be considered statistically significant. The analysis will be performed with the STATA16 statistical package. 

### 2.11. Participation of Subjects with HFpEF in Research

The design and purpose of the RCT will be exposed to an advisory group of subjects with HF-PEF from the Cuenca I, II and III health centers of Cuenca. This advisory group will meet periodically during the duration of the study, especially for the drafting of clinical implications in the reports and conclusions. In addition, they will participate in the design of dissemination material after being informed of the findings and conclusions and will contribute to the dissemination plan.

## 3. Results

The results will be published as a peer-reviewed article.

## 4. Discussion

The primary objective of this study will be to compare the effectiveness of CE and HIIT on exercise capacity, diastolic function, endothelial function, and arterial stiffness in participants with HFpEF, and the secondary objective of this study will be to compare the effectiveness CE and HIIT on QoL in participants with HFpEF.

### 4.1. HFpEF and Exercise Capacity

One of the hallmarks of HFpEF is severe impairment of exercise capacity. Exercise intolerance, manifested by symptoms of exertional dyspnea and fatigue, impairs QoL and is therefore one of the main consequences for participants with HFpEF. Furthermore, reduced exercise capacity in HFpEF is associated with worse clinical outcomes [36]. Studies to date have used a variety of techniques to examine the mechanisms of exercise intolerance in HFpEF and have variably reported the contribution of central (e.g., HR, stroke volume, filling pressures) and peripheral (e.g., VO2max, body mass index, renal function) factors.

### 4.2. HFpEF and Diastolic Function

HFpEF is characterized by abnormal diastolic function: there is an increase in left ventricular stiffness, leading to decreased left ventricular relaxation during diastole, with a consequent increase in pressure and/or impaired filling. There is evidence that long-term aerobic exercise may improve diastolic function because this exercise attenuates left ventricular remodeling due to a reduction in vasoconstrictor neurohormones or a decrease in hemodynamic load [17]. However, this evidence has not been demonstrated in participants with HFpEF, nor has it been analyzed with other types of exercise.

### 4.3. HFpEF and Endothelial Function

The relationship between endothelial dysfunction and HF has been well established in many studies [37]. Endothelial function is not only closely associated with HFpEF but also strongly predicts events in participants with HFpEF [38]. This is because endothelial cells (1) participate in antioxidant and anti-inflammatory activities in arteries; (2) interact with the extracellular matrix (elastic and collagen fibers) to regulate vascular elasticity; and (3) directly affect vascular tone by synthesizing and releasing nitric oxide [39]. Endothelial function plays an important role in arterial stiffness and HFpEF.

### 4.4. HFpEF and Arterial Stiffness

The change in arterial hemodynamics plays a key role in HFpEF. Arterial stiffness is a determinant of pulsatile afterload, which is one of the two main components of arterial load. Arterial wave reflections increase with arterial stiffness, leading to increased mid and late systolic load and subsequent left ventricular abnormalities, such as concentric remodeling, myocardial fibrosis, contractile dysfunction, and reduced ejection duration. These changes further contribute to the increase in mid and late systolic load, resulting in a vicious cycle [40]. Because of the contribution of arterial stiffness in HFpEF, therapies targeting arterial stiffness may be useful in the treatment of this disease. To date, aerobic exercise, CE and HIIT have been shown to be the most effective in reducing arterial stiffness [41]; however, these findings have not yet been demonstrated in subjects with HFpEF.

### 4.5. HFpEF and QoL

Given the paucity of treatments that decrease mortality in HFpEF, there is interest in focusing on possible treatments to improve QoL in these patients [42], taking into account the multidimensional concept of QoL (physical and psychological dimensions) [43]. Patients with HFpEF are usually elderly and their main chronic symptom is severe exercise intolerance which reduces their QoL [16]. Functional capacity and activities of daily living correlate more strongly with QoL impairment in this type of patient [42], showing a lower QoL in the physical dimension compared with other chronic patients [43]. Previous meta-analyses and several ECAs have shown an improvement in exercise capacity and QoL after training in patients with HF [26,43,44].

## 5. Conclusions

With this study, we will provide evidence to healthcare professionals on the efficacy of CE and HIIT in improving exercise capacity, diastolic function, endothelial function and arterial stiffness in participants with HFpEF, as well as improving QoL, considered as a therapeutic goal for these patients. Furthermore, taking into account the increasing prevalence of participants with HF, it is plausible that the clinical importance of HF will increase in the coming years. Therefore, the management of HF is a major challenge for healthcare systems, where it would be important to consider the prescription of physical exercise as a complementary nonpharmacological treatment in this type of patient.

## Figures and Tables

**Figure 1 jcm-12-03535-f001:**
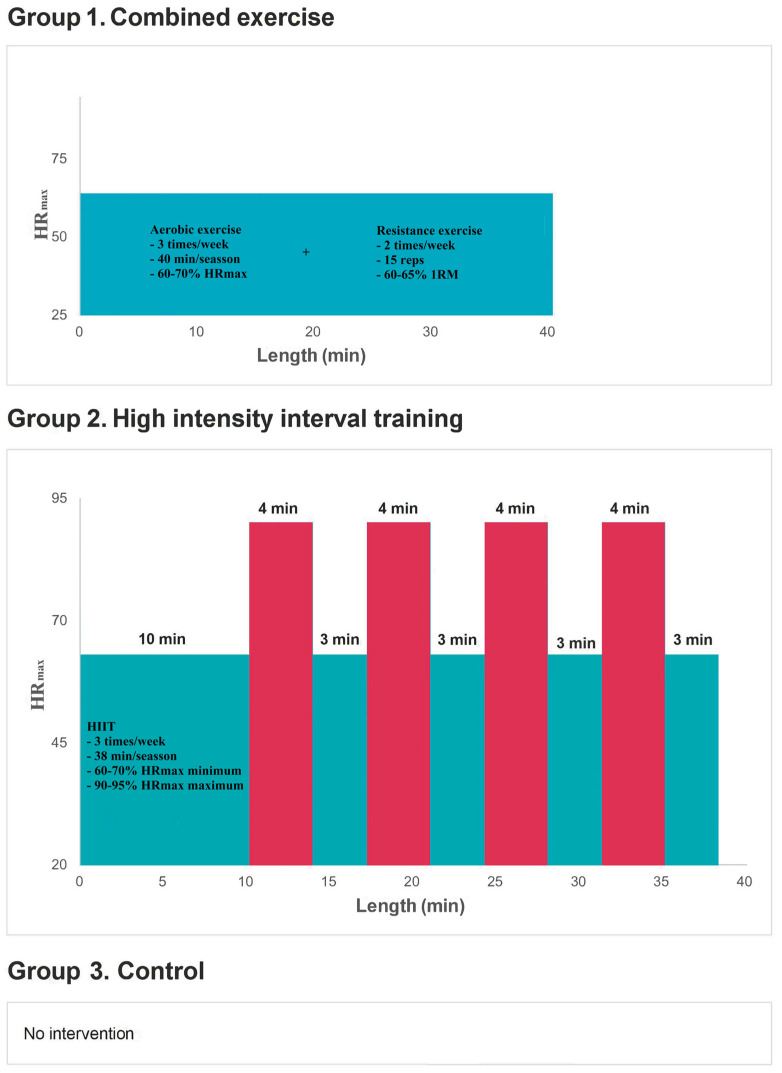
Combined exercise (group 1), high intensity interval training (group 2) and control group protocols (group 3).

**Figure 2 jcm-12-03535-f002:**
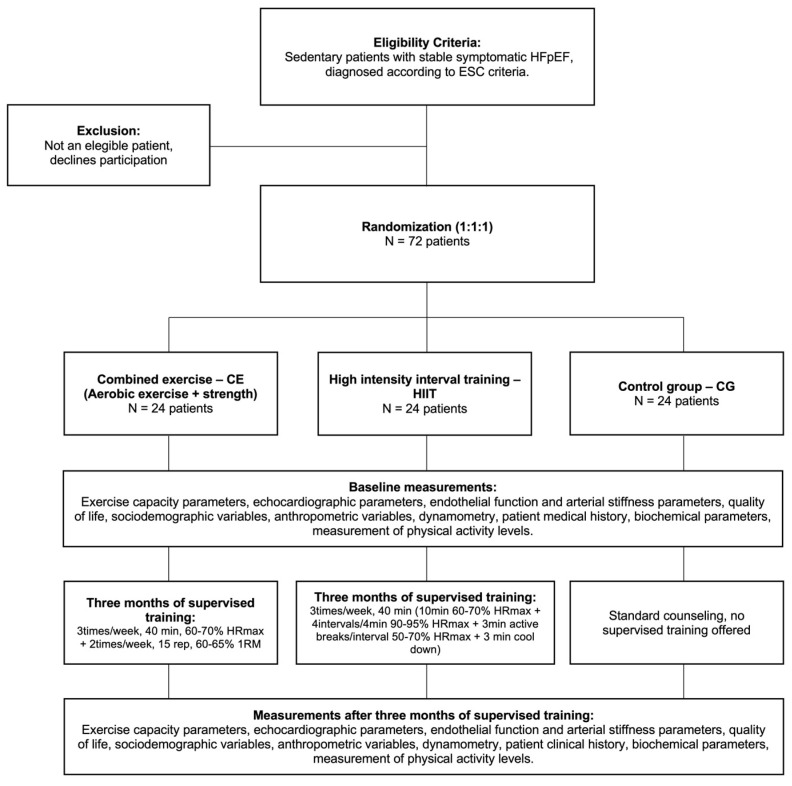
Flow chart for participant inclusion and follow-up: screening, inclusion and exclusion, randomization, baseline testing, and testing after follow-up.

**Table 1 jcm-12-03535-t001:** Inclusion and exclusion criteria.

Inclusion Criteria	Exclusion Criteria
1. HFpEF (diagnosis according to ESC 2021 criteria): a. Signs and symptoms of HF. b. LVEF ≥ 50%. c. Objective evidence of cardiac structural and/or functional abnormalities consistent with the presence of left ventricular diastolic dysfunction/elevated left ventricular filling pressures, including elevated natriuretic peptide. 2. Sedentary men and women (structured exercise < 2 × 30 min/week). 3. Age ≥ 40 years. 4. Written informed consent. 5. Clinically stable for ≥6 weeks. 6. Optimal medical treatment for ≥6 weeks.	1. Noncardiac causes of HF symptoms: -Significant valvular or coronary artery disease. -Uncontrolled hypertension or arrhythmias. -Primary cardiomyopathies. 2. Significant pulmonary disease (FEV1 < 50% predicted, GOLD III-IV). 3. Exercise disability or conditions that may interfere with exercise intervention. 4. Myocardial infarction within the last 3 months. 5. Signs of ischemia during maximal cardiopulmonary stress test. 6. Comorbidity (renal failure, cancer, cognitive impariment) that may influence prognosis at 1 year. 7. Participation in another clinical trial.

ESC, European Society of Cardiology; HF, heart failure; HFpEF, heart failure with preserved ejection fraction; LVEF, left ventricular ejection fraction; GOLD, Global Initiative for Chronic Obstructive Lung Disease; FEV1, forced expiratory volume in the first second.

## Data Availability

Not applicable.

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
