# Peer review of "Comparative Effect of Two Types of Physical Exercise for the Improvement of Exercise Capacity, Diastolic Function, Endothelial Function and Arterial Stiffness in Participants with Heart Failure with Preserved Ejection Fraction (ExIC-FEp Study): Protocol for a Randomized Controlled Trial"

_jcm, 2023, doi:10.3390/jcm12103535_

Round 1
Reviewer 1 Report
The idea of this future study is very original and if it could be developed exactly it is written, this controlled trial will be a high quality study.
- Line 40: In text 3, they say that the survival of patients with heart failure with preserved ejection fraction was similar to that of patients with reduced ejection fraction, not that the prognosis of HFpEF is almost comparable to HFrEF. In generally, HFrEF has a worse prognosis. This include deaths, hospitalization, quality of life...
- Line 103 / Table 1: Exclusion criteria
6. Comorbidity that may influence prognosis at 1 yearàyou may describe these comorbidities, like cancer, smoking, diabetes complications
- Line 167: These large number of outcomes may difficult the data analyzes and the conclusions. Think about this.
- Line 319: describe better what does ¨quality of life¨ means
Author Response
Reviewer 1
The idea of this future study is very original and if it could be developed exactly it is written, this controlled trial will be a high quality study.
Authors:
Thank you for the reviewer’s comment. We greatly appreciate the time the reviewer spent reviewing the manuscript.
- Line 40: In text 3, they say that the survival of patients with heart failure with preserved ejection fraction was similar to that of patients with reduced ejection fraction, not that the prognosis of HFpEF is almost comparable to HFrEF. In generally, HFrEF has a worse prognosis. This include deaths, hospitalization, quality of life...
Authors:
The reviewer´s comment seems judicious. As suggested, we have modified this issue as follows:
“[…] Generally, the prognosis of HFpEF is slightly better than the prognosis of HFrEF [3,4]. […]”
- Line 103 / Table 1: Exclusion criteria
- Comorbidity that may influence prognosis at 1 year you may describe these comorbidities, like cancer, smoking, diabetes complications.
Authors:
Thank you for the reviewer’s comment. As suggested, we have described the comorbidities.
“[…] 6. Comorbidity (renal failure, cancer, cognitive impairment) that may influence prognosis at 1 year. […]”
- Line 167: These large number of outcomes may difficult the data analyzes and the conclusions. Think about this.
Authors:
Thank you for the reviewer’s comment. The ExIC-FEp study is a funded study in which different main outcomes such as exercise capacity, diastolic function, endothelial function and arterial stiffness will be measured, but when analyzing the data, each of the main outcomes will be evaluated separately, to facilitate the analysis of the data and to be able to draw specific conclusions on the effect of two types of exercise for each of the outcomes.
- Line 319: describe better what does ¨quality of life¨ means.
Authors:
The reviewer´s comment seems judicious. As suggested, we have described the concept of ¨quality of life¨ throughout the manuscript.

Reviewer 2 Report
Certainly, it’s an interesting topic, considering the lack of works focusing on comparing these two types of training.
In literature we can found several meta-analyses and reviews on the impact of physical exercise in optimize exercise capacity and reducing cardiovascular risk factors in patients with heart failure. The reported analysis of previous publications on this subject is accurate; I have not found other paper that need to be cited.
As I said before, I agree that focusing also on the endothelial response and comparing combined training with high intensity interval training in patients with heart failure with preserved ejection fraction, it could certainly be an interesting RCT.
However, I have some queries and suggestion that can be observed.
First, it could be useful to better explain modality of acquisition of echocardiographic values use to recognize inclusion criteria, as well as methods used for exclusion criteria.
Then, I have got a major concern about the explanation of secondary endpoint: a focus on quality of life comes over only in the conclusions. Maybe there could be more space describing this point.
Table and pictures are well structured, even if a table of basic clinical feature is missing.
Talking about statistical analysis, no particular concern can be found.
Discussion and conclusion seem reasonable and well referred, even without reported results.
With regards to minor points, the article is well written, without major grammatical errors.
Good luck with your paper and thank you again for your submission.
Author Response
Reviewer 2
Certainly, it’s an interesting topic, considering the lack of works focusing on comparing these two types of training.
In literature we can found several meta-analyses and reviews on the impact of physical exercise in optimize exercise capacity and reducing cardiovascular risk factors in patients with heart failure. The reported analysis of previous publications on this subject is accurate; I have not found other paper that need to be cited.
As I said before, I agree that focusing also on the endothelial response and comparing combined training with high intensity interval training in patients with heart failure with preserved ejection fraction, it could certainly be an interesting RCT.
However, I have some queries and suggestion that can be observed.
Authors:
Thank you for the reviewer’s comment. We greatly appreciate the time the reviewer spent reviewing the manuscript.
First, it could be useful to better explain modality of acquisition of echocardiographic values use to recognize inclusion criteria, as well as methods used for exclusion criteria.
Authors:
The reviewer´s comment seems judicious. As suggested, we have included the following information regarding the establishment of inclusion and exclusion criteria:
“For inclusion and exclusion criteria, a complete echocardiographic examination incorporating all two-dimensional and Doppler data will be performed. In addition, the patient's clinical assessment, exercise capacity and medical history will be taken into account and spirometry will be performed to rule out significant pulmonary disease.”
Then, I have got a major concern about the explanation of secondary endpoint: a focus on quality of life comes over only in the conclusions. Maybe there could be more space describing this point.
Authors:
The reviewer´s comment seems judicious. As suggested, we have described the concept of ¨quality of life¨ throughout the manuscript.
Table and pictures are well structured, even if a table of basic clinical feature is missing.
Authors:
Thank you for the reviewer’s comment. As suggested, we have included the following table of basic clinical feature in the manuscript:
“The basic clinical features are shown in Table S1.”
Table 1. Basic clinical features.
Primary outcomes |
Covariates |
|||||||||||
Exercise capacity |
Echocardiography |
Endothelial function |
Arterial stiffness |
Quality of life |
Sociodemographic variables |
Patient medical history |
Anthropometric variables |
Muscle strength |
Spirometry |
Biochemical parameters |
Physical activity |
Mediterranean diet adherence |
VO2max (ml/min/kg)
Workload (watts)
Exercise time (s)
HRmax (beats/min)
6-minute walking test (m) |
E velocity (m/s)
A velocity (m/s)
E/A ratio
e' velocity ratio
Ejection fraction percentage (%)
Left ventricular volume index (mL/m2)
End-diastolic volume (mL)
Left ventricular mass (g)
Left atrial diameter (mm)
Isovolume relaxation time (ms)
Deceleration time (ms)
Left atrial volume index
|
cIMT (mm) |
PWv (m/s)
cAIx (%) |
SF-12
MLWHFQ |
Age (years)
Sex
Socioeconomic level |
Comorbidities (n, %)
Pharmacological treatment |
Weight (kg)
Height (cm)
BMI (kg/m2)
Densitometry
Blood pressure (mmHg) |
Handgrip strength (kg) |
FVC (L)
FEVI (L)
FEVI/FVC (%) |
Glucose (mg/dL)
Total cholesterol (mg/dL)
Triglycerides (mg/dL)
HDL cholesterol (mg/dL)
LDL cholesterol (mg/dL)
Apolipoproteins A1 and B (mg/dL)
Insulin (mlU/L)
Ultrasensitive C-reactive protein (mg/L)
NT-proBNP (pg/mL)
HbA1c (%) |
Accelerometry (g)
Podometry (steps/day) |
MEDAS-14 |
Talking about statistical analysis, no particular concern can be found.
Authors:
Thank you for the reviewer’s comment.
Discussion and conclusion seem reasonable and well referred, even without reported results.
Authors:
Thank you for the reviewer’s comment.
With regards to minor points, the article is well written, without major grammatical errors.
Good luck with your paper and thank you again for your submission.
Authors:
Thank you for the reviewer’s comment.

Reviewer 3 Report
The effect of physical exercise on HF was studied. This is a very good study. The logical starting point, method and process of the study are very appropriate. However, the conclusion part of the article does not completely present the research process. It is suggested that the conclusion part should improve the findings of physical exercise on HF in this paper, and also enrich the practical guidance of physical exercise on HF. References can also be appropriately updated to point to the latest research.
Author Response
Reviewer 3
The effect of physical exercise on HF was studied. This is a very good study. The logical starting point, method and process of the study are very appropriate.
Authors:
Thank you for the reviewer’s comment. We greatly appreciate the time the reviewer spent reviewing the manuscript.
However, the conclusion part of the article does not completely present the research process. It is suggested that the conclusion part should improve the findings of physical exercise on HF in this paper, and also enrich the practical guidance of physical exercise on HF.
Authors:
Thank you for the reviewer’s comment. As suggested, we have modified this issue as follows:
“With this study, we will provide evidence to healthcare professionals on the efficacy of CE and HIIT in improving exercise capacity, diastolic function, endothelial function and arterial stiffness in participants with HFpEF, as well as improving QoL, considered as a therapeutic goal for these patients. Furthermore, taking into account the increasing prevalence of participants with HF, it is plausible that the clinical importance of HF will increase in the coming years. Therefore, the management of HF is a major challenge for healthcare systems, where it would be important to consider the prescription of physical exercise as a complementary nonpharmacological treatment in this type of patients.”
References can also be appropriately updated to point to the latest research.
Authors:
Thank you for the reviewer’s comment. As suggested, we have updated the following references to point to the latest research.
“3. KapÅ‚on-CieÅ›licka, A.; KupczyÅ„ska, K.; Dobrowolski, P.; Michalski, B.; Jaguszewski, M. J.; Banasiak, W.; et al. On the search for the right definition of heart failure with preserved ejection fraction. Cardiol J 2020, 27(5), 449-468.
11. Heidenreich, P. A.; Bozkurt, B.; Aguilar, D.; Allen, L. A.; Byun, J. J.; Colvin, M. M.; et al. 2022 AHA/ACC/HFSA Guideline for the Management of Heart Failure: Executive Summary: A Report of the American College of Cardiology/American Heart Association Joint Committee on Clinical Practice Guidelines. J Am Coll Cardiol 2022, 79(17), 1757-1780.
42. Reddy, Y. N. V.; Rikhi, A.; Obokata, M.; Shah, S. J.; Lewis, G. D.; AbouEzzedine, O. F.; et al. Quality of life in heart failure with preserved ejection fraction: importance of obesity, functional capacity, and physical inactivity. Eur J Heart Fail 2020, 22(6), 1009-1018.
43. Moradi, M.; Daneshi, F.; Behzadmehr, R.; Rafiemanesh, H.; Bouya, S.; Raeisi, M. Quality of life of chronic heart failure patients: a systematic review and meta-analysis. Heart Fail rev 2020, 25(6), 993–1006.
44. Fu, T. C.; Yang, N.I.; Wang, C.H.; Cherng, W. J.; Chou, S. L.; Pan, T. L.; et al. Aerobic interval training elicits different hemodynamic adaptations between heart failure patients with preserved and reduced ejection fraction. Am J Phys Med Rehabil 2016, 95, 15-27.”
